# Differences in Clinical Features among Different Onset Patterns in Moyamoya Disease

**DOI:** 10.3390/jcm10132815

**Published:** 2021-06-25

**Authors:** Yudai Hirano, Satoru Miyawaki, Hideaki Imai, Hiroki Hongo, Yu Teranishi, Shogo Dofuku, Daiichiro Ishigami, Kenta Ohara, Satoshi Koizumi, Hideaki Ono, Hirofumi Nakatomi, Nobuhito Saito

**Affiliations:** 1Department of Neurosurgery, The University of Tokyo, Bunkyo-ku, Tokyo 113-8655, Japan; yudai.hrn@gmail.com (Y.H.); hiimai-nsu@umin.ac.jp (H.I.); hhongo@hotmail.co.jp (H.H.); eugeneterra0725@gmail.com (Y.T.); s.dofuku@gmail.com (S.D.); speedm217@gmail.com (D.I.); ohara.kenta.729@gmail.com (K.O.); koizumisatoshi@gmail.com (S.K.); hideono-tky@umin.ac.jp (H.O.); hnakatomi-tky@umin.ac.jp (H.N.); nsaito-nsu@m.u-tokyo.ac.jp (N.S.); 2Department of Neurosurgery, Fuji Brain Institute and Hospital Fujinomiya, Shizuoka 418-0021, Japan; 3Department of Neurosurgery, Japan Community Healthcare Organization Tokyo Shinjuku Medical Center, Shinjuku-ku, Tokyo 162-8543, Japan; 4Department of Neurosurgery, Kyorin University Hospital, Mitaka, Tokyo 181-8611, Japan

**Keywords:** asymptomatic moyamoya disease, cerebrovascular events, hemorrhagic onset, ischemic onset, moyamoya disease, natural history, onset pattern, risk factors

## Abstract

Moyamoya disease is characterized by severe stenosis at the ends of the bilateral internal carotid arteries and the development of collateral circulation. The disease is very diverse in terms of age at onset, onset patterns, radiological findings, and genetic phenotypes. The pattern of onset is mainly divided into ischemic and hemorrhagic onsets. Recently, the opportunity to identify asymptomatic moyamoya disease, which sometimes manifests as nonspecific symptoms such as headache and dizziness, through screening with magnetic resonance imaging has been increasing. Various recent reports have investigated the associations between the clinical features of different onset patterns of moyamoya disease and the corresponding imaging characteristics. In this article, we have reviewed the natural history, clinical features, and imaging features of each onset pattern of moyamoya disease.

## 1. Introduction

The main pathology of moyamoya disease is chronic progressive stenosis at the terminal portions of the bilateral internal carotid arteries and the development of abnormal collateral circulation, the so-called “moyamoya vessels” around the base of the brain [1]. The incidence of moyamoya disease is particularly high in East Asian populations such as those in Japan, Korea, and China, and lower in populations in other areas [2]. Moyamoya disease is also characterized by the diversity of disease phenotypes, such as age of onset, onset pattern, imaging findings, and genetic mutations. The onset patterns are mainly divided into ischemic onset, such as transient ischemic attacks (TIAs), and cerebral infarction, and hemorrhagic onset, such as intracerebral hemorrhage and subarachnoid hemorrhage. During childhood, the disease most often manifests as ischemic symptoms, while in adults, it may sometimes present as cerebral hemorrhage in addition to cerebral ischemia.

There are still many unclear points regarding the factors that determine the type of onset. Recent studies have revealed the clinical features and imaging characteristics of each onset pattern. For example, the associations between angiographic characteristics of periventricular anastomosis and hemorrhagic onset have been investigated [3]. There is also one report comparing each clinical feature, including age of onset, stroke risk factors, family history, and radiological features, in the asymptomatic, ischemic, and hemorrhagic groups [4].

With the widespread use of non-invasive diagnostic imaging methods such as magnetic resonance imaging (MRI), the possibility of diagnosis before the onset of moyamoya disease is increasing. However, there are fewer accurate epidemiological data on asymptomatic moyamoya disease. In addition, it is not clear what risk factors cause cerebrovascular events in asymptomatic cases. Limited large-scale studies have been conducted on asymptomatic moyamoya disease, and little is known about its natural history. If it becomes clear what risk factors cause cerebrovascular events in asymptomatic cases, it may be possible to prevent symptomization and improve outcomes.

In this article, we review the research on the natural history, clinical features, and imaging features of each onset pattern of moyamoya disease based on past reports, including recently published studies.

## 2. Purpose and Methods

The purpose of this study was to review the clinical and radiological characteristics of moyamoya disease by onset patterns. Based on the results of this study, it is highly possible that future cerebrovascular events can be prevented by medical or surgical intervention in appropriate cases. As a selection criterion for the study, we searched Pubmed with the search formula “(moyamoya [Title]) AND((risk) OR (predictor) OR (frequent)) AND ((ischemic) OR (asymptomatic) OR (hemorrhagic)) AND ((onset) OR (presentation) OR (natural history))”. Among the results of searching with the formula, we focused on papers that analyzed clinical or radiological factors associated with each onset pattern and symptomization in asymptomatic moyamoya disease. We included case series, cohort studies, and reviews that reported the factors associated with the onset pattern of moyamoya disease. We excluded studies only about the surgical results, case reports, and articles not written in English. Moyamoya disease has a high prevalence in Asia and a relatively low prevalence in the United States and Europe. Therefore, although epidemiological studies are scattered, there is no large cohort classified by the onset pattern in Caucasian populations. In this review as well, many Asian cohorts are selected.

## 3. Clinical Factors Associated with Ischemic Onset Moyamoya Disease

In moyamoya disease, as the stenosis of the end of the bilateral carotid arteries progresses, cerebral blood flow is usually compensated for by the development of collateral circulation, called moyamoya vessels. As a natural course, the source of cerebral blood flow gradually shifts from the internal carotid artery to the external carotid/vertebral artery [5]. The compensatory system fails when the collateral circulation cannot develop in time due to the rapid progression of stenosis in the internal carotid artery system or when complicated with stenosis of the posterior cerebral artery. Breakdown of the compensatory mechanism reduces cerebral blood flow, which causes hemodynamic TIAs or ischemic strokes. Hyperventilation, such as when crying or blowing a whistle, induces ischemic attacks, which causes cerebral vasospasm.

Epidemiologically, the incidence of cerebral ischemia associated with moyamoya disease was 0.53 patients per 100,000 people-years in Japan, which showed two peaks at 5–9 years and 45–49 years [6]. In China, the incidence of ischemia associated with the disease was 0.16 patients per 100,000 people-years, which showed two peaks at 5–9 years and 35–39 years [7]. Bimodal peaks have also been observed in moyamoya patients in the United States [8]. In adults, approximately half of symptomatic moyamoya disease patients develop ischemia [2,9]. The anterior circulatory system, which includes the frontal, parietal, and temporal lobes, is most likely to be involved, causing focal neurological signs such as dysarthria, aphasia, hemiparesis, and cognitive impairment [10]. Patients may also have seizures, visual deficits, syncope, or personality changes that can be mistaken for psychiatric illnesses [11]. Ischemic onset is the most common early symptom of moyamoya disease in children. Cerebral ischemia is observed in 78% of patients under the age of 10 years compared to 53.5% of patients aged 10 years or more [6]. Younger infants are more likely to have an ischemic stroke, and the presence of ischemia is most relevant to functional prognosis [12,13]. Of the patients with moyamoya disease who are under 3 years of age, 87% develop cerebral infarction, and 39% relapse in a short period of time [13]. Intellectual disabilities and neurological deficits worsen with the passage of time from onset, but stabilize in about 10 years [14,15]. In addition, the disease advances in angiographic stages between childhood and adolescence, but stabilizes between adolescence and adulthood [16]. Based on such past data, it is considered that the stage progresses and the hemodynamic status becomes unstable within a few years after the appearance of ischemic symptoms; thus, aggressive revascularization is desirable. Furthermore, 65% of children with moyamoya disease who had unilateral ischemic symptoms had TIAs on the asymptomatic side [17]. Therefore, careful follow-up is required for the asymptomatic side as well.

Various factors have been proposed as predictors of ischemic onset in moyamoya disease. These can be divided into clinical features and imaging characteristics. The first clinical feature, as mentioned above, is that younger children have a higher rate of cerebral infarction as an initial presentation. Infarctions are significantly more frequent in children aged <3 years (87%) and 3–6 years (58%) than in those aged 6 years or more (46%) [13]. TIAs have a relatively high recurrence rate in children with moyamoya disease and are also associated with subsequent ischemic stroke. In pediatric moyamoya disease, female sex is an independent factor that predicts the progression from TIA to cerebral infarction [18]. In a report analyzing the background factors for 71 patients with ischemia and adult moyamoya disease, there was no significant difference in the proportion of women, stroke risk factors, and family history between the symptomatic and asymptomatic groups [4].

Moyamoya disease has a significant genetic background. In a retrospective study on 241 patients with moyamoya disease, multivariate analysis of risk factors in the ischemic onset group revealed that familial moyamoya disease was a significant factor [19]. In recent years, the *RNF213* gene has been identified as a disease susceptibility gene and 80% of patients with moyamoya disease have *RNF213* c.14429G>A (p.Arg4810Lys, rs112735431) (based on the National Center for Biotechnology Information Reference Sequences NM_001256071 and NP_00124300), which is the most frequent gene variant observed [20,21]. Recent studies have suggested that *RNF 213* polymorphisms are associated with clinical phenotypes. The presence of the *RNF213* c.14429G>A (p.Arg4810Lys, rs112735431) mutation in patients with moyamoya disease has been implicated in earlier onset, increased risk of cerebral infarction as an initial presentation, and more frequent posterior cerebral artery (PCA) involvement [22,23,24,25,26,27,28,29].

The relationship between specific thyroid condition and moyamoya disease has been reported for some time, and it has also been reported to be related to clinical manifestation. Specifically, hyperthyroidism and elevated anti-thyroperoxidase antibody are likely to be associated with ischemic stroke in adult moyamoya disease [30].

In terms of radiological characteristics, factors that predict cerebral infarction have also been clarified. In a report analyzing the predictors of subsequent stroke in 60 pediatric patients with moyamoya disease who presented with TIA, Suzuki’s angiographical stage greater than 3 was an independent predictor of future strokes [18]. Cerebral infarction in the posterior circulation was considered uncommon, but recent studies have shown that PCA involvement was present in approximately 33% of adult patients and 26% of pediatric patients [31]. The prevalence of infarction in patients with PCA involvement was significantly higher than that in patients without PCA involvement in both pediatric and adult patients [32]. In addition, the frequency of ipsilateral cerebral infarction was significantly correlated with the advancement of PCA stages in adult patients, but not in pediatric patients [33]. Furthermore, occlusive changes in the proximal carotid artery are associated with ischemic moyamoya disease [32]. The specific imaging findings of vessel wall imaging using MRI with a contrast agent may also be significantly related to ischemic onset. Strong enhancement of the intracranial vessel wall was found to be associated with the progression of intracranial artery stenosis, while a lack of enhancement was associated with stability of intracranial artery stenosis [34]. As a clinical relevance, the presence of contrast enhancement and wall thickening showed a statistically significant association with ischemic events [35]. It has also been suggested that the modality of blood flow evaluation suggests an underlying condition for the ischemic onset. Hemodynamic parameters such as relative cerebral blood volume (CBV) and relative mean transit time (MTT) are measured using dynamic susceptibility contrast MRI in adult patients. In one study, the relative value of CBV and prolongation of the MTT were significantly higher in the infarction and TIA groups than in the asymptomatic group [36]. In pediatric patients with moyamoya disease, measuring blood oxygen level-dependent cerebrovascular reactivity using MRI could be predictive of ischemic events. The presence of steal and idiopathic moyamoya was found to be independently associated with ischemic events in another study [37]. The relative cerebral blood flow evaluated by single-photon emission computed tomography (SPECT) detects misery perfusion with high sensitivity [38]. The ratio of cerebral blood flow in one lenticular nucleus to that in the peripheral middle cerebral artery (MCA) area which is called hemodynamic stress distribution is associated with ischemic symptoms, while cerebral blood flow at rest in the lenticular nucleus or MCA is not significantly associated with clinical manifestation. More specifically, the hemodynamic stress distribution was 1.08 in the ischemic group, whereas it was 1.03 in the asymptomatic group. In addition, ROC analysis that was performed to set the optimal threshold revealed that patients tended to exhibit ischemic symptoms at a hemodynamic stress distribution 1.04 or higher [39]. In addition, microembolic signals detected by transcranial Doppler ultrasonography are also associated with ischemic moyamoya disease [40,41]. Clinical factors associated with ischemic onset moyamoya disease are summarized in a table (Table 1).

## 4. Factors Associated with Recurrence of Ischemic Symptoms in Ischemic Onset Moyamoya Disease

Ischemic events have a high recurrence rate when treated conservatively. It has been reported that the same type of stroke as that at initial presentation is likely to occur in adult patients with moyamoya disease. In a retrospective analysis of 241 patients with moyamoya disease, a multivariate analysis of risk factors revealed that initial ischemic presentation was a significant factor for ischemic events [19]. In children, the recurrence rate was inversely correlated with age, and was as high as 39% in children under 3 years of age [13]. In patients treated non-surgically, the rate of ischemic stroke recurrence was 1.6% in the first year and 11.8% in the fifth year [42]. According to a study that analyzed the risk of recurrence of ischemic moyamoya disease, when the course was followed by conservative treatment, diabetes mellitus and PCA stenosis, decreased cerebrovascular reserve (CVR) as radiological factors were identified as predictors of ischemic stroke recurrence [42]. A decrease in CVR suggests a collapse of the compensatory mechanism, and PCA stenosis causing a decrease in blood flow due to poor compensation for leptomeningeal anastomosis from the posterior circulation [43]. The ivy sign on MRI, defined as a linear or focal high-signal intensity on fluid-attenuated inversion recovery images, is also a predictor of cerebral infarction recurrence in adult patients with moyamoya disease. Of the 154 hemispheres of 84 patients analyzed in one study, cerebral infarction recurred in 9 hemispheres within 3 years, and the ipsilateral ivy sign was found to be an independent risk factor for recurrent ischemic stroke in multivariate analysis [44]. Factors associated with recurrence of ischemic symptoms in ischemic onset moyamoya disease are summarized in a table (Table 2).

## 5. Factors Associated with Hemorrhagic Transformation in Ischemic Onset Moyamoya Disease

In some cases, ischemic onset moyamoya disease can undergo hemorrhagic transformation. In one study analyzing 683 patients with ischemic onset moyamoya disease, 29 (4.3%) cases were of this type; moreover, multivariate analysis revealed that normal cerebral perfusion was related to hemorrhagic transformation. Hemorrhagic transformation often has worse outcomes than the normal course of the disease, and should be noted [45]. It has been reported that cerebrovascular events such as hemorrhagic transformation occur in patients who had undergone bypass surgery in childhood [46]—of the fifty-six patients (follow-up period, 18.1 years on average), and four patients had delayed cerebrovascular events. Of these, hemorrhagic complications occurred in three cases, with an annual incidence of 0.3%. All cases occurred in patients above 20 years of age, and more than 10 years after the surgery. In addition, it has been reported that the probability of late-onset cerebral hemorrhage in pediatric moyamoya disease is approximately 0.4% for a follow-up period of less than 10 years, but 3.1% for follow-up periods of 10 years or more [47], which indicates that long-term follow-up is necessary. Factors associated with hemorrhagic transformation in ischemic onset moyamoya disease are summarized in a table (Table 3).

## 6. Clinical Factors Associated with Hemorrhagic Onset Moyamoya Disease

In moyamoya disease, as the stenosis of large blood vessels progresses, the load on the collateral circulation becomes excessive. The pathogenesis of hemorrhagic moyamoya disease is presumed to be the rupture of abnormal collateral vessels due to this hemodynamic load or of an aneurysm formed in the circle of Willis.

Epidemiologically, the incidence of hemorrhage associated with moyamoya disease was 0.2 patients in Japan and 0.22 patients in China per 100,000 people-years [6,7]. However, the incidence varies depending on the region, and the incidence of hemorrhage in Asian adults is much higher than that in adults in the United States [48,49]. Furthermore, the frequency of hemorrhagic events is significantly lower in European moyamoya disease [50]. Therefore, there is no integrated report analyzing the risk factors for hemorrhagic onset in patients with moyamoya disease in Europe and the United States. Hemorrhagic events are common in adult patients, with a peak at 35–39 years [6,7] or 30–34 years [51]. In the United States, the age of patients hospitalized for bleeding generally peaks in the 1960s and 1970s [8]. In addition, the incidence of hemorrhage in pediatric patients is only about 3% in Asian countries, which means that it is a rare clinical manifestation in children [52]. The most frequent sites of bleeding in moyamoya disease are intraventricular, lobar, and putaminal [53]. Hemorrhagic onset is the most important factor that worsens the prognosis of moyamoya disease [48]. The prognosis of hemorrhagic moyamoya disease is poor, and the mortality rate due to initial intracranial hemorrhage is approximately 6.8–18% [54,55]. Severe ischemic stroke may occur in patients with hemorrhagic stroke if the hemodynamic reserve is impaired [56].

Regarding the factors associated with hemorrhagic onset, sex, frequency of stroke risk factors, and family history of hemorrhagic moyamoya disease were not significantly different from those for other onset types such as asymptomatic or ischemic moyamoya disease [4]. Adult moyamoya disease with childhood onset is associated with a hemorrhagic onset. A multivariate analysis of 69 untreated patients divided into childhood onset and adulthood onset groups according to the onset age showed that childhood onset adult moyamoya disease was significantly associated with hemorrhagic moyamoya disease [57]. In a recent report evaluating periventricular anastomosis on angiography in pediatric ischemic, adult ischemic, and adult hemorrhagic groups, the development of collateral circulation was more prevalent in the pediatric ischemic and adult hemorrhagic groups than in the adult ischemic group [58]. This suggests that collateral circulation developed more frequently in the childhood-onset group, which causes future bleeding. As for stroke-related risk factors, hypertension is also reported to be a risk factor for anterior hemorrhage in patients without abnormal dilation of choroidal anastomosis [59]. Familial moyamoya disease has also been reported as a significant risk factor for hemorrhagic events [19].

There have been several reports on the relationship between imaging characteristics and hemorrhagic moyamoya disease in recent years. Suzuki’s angiographic stages are found to be significantly more advanced in the hemorrhagic group than in the ischemic group or asymptomatic group [60,61,62]. This phenomenon is observed not only in the Asian cohort but also in the American cohort [63]. Moreover, occlusions of the anterior cerebral artery, development of fetal-type PCA, and intracranial aneurysms were more frequent in the hemorrhagic hemispheres in adult patients, indicating that these angiographic features are contributing factors to hemorrhagic onset [61]. Angiographic findings of dilatation and branch extension of the anterior choroidal artery, posterior choroidal artery, posterior communicating artery, and medullary arteries in adult patients are also associated with hemorrhagic onset [64,65,66,67]. Marked development of the anterior choroidal artery detected using MR angiography (MRA), which is less invasive than cerebral angiography, is also associated with hemorrhagic onset [4]. In addition, it has been suggested that periventricular anastomosis detected by MRA is associated with the hemorrhagic onset [68]. More detailed studies using cerebral angiography have since been reported, which showed that patients with hemorrhagic onset showed a significantly higher proportion of thalamic anastomosis and choroidal anastomosis compared to asymptomatic patients or patients with ischemic onset [60,62]. Hemorrhage can occur in the anterior or posterior cerebral region, depending on the site of bleeding, and it has been reported that choroidal anastomosis is a characteristic of posterior hemorrhage [3,59]. Like choroidal anastomosis, PCA involvement is more prevalent in the hemorrhagic group than in the asymptomatic group [62] and is associated with posterior hemorrhage [3,59]. The finding of microbleeds on MRI T2* images is associated with hemorrhage (present in approximately 43% of hemorrhagic onset patients) [4]. The presence of multiple microbleeds is an independent risk factor for subsequent hemorrhage [69]. It has also been suggested that the site of microbleeds is associated with hemorrhagic onset, and that the presence of microbleeds in the deep and periventricular white matter is significantly associated with subsequent intraventricular hemorrhage [70]. The frequency of cerebral microbleeds in European patients with moyamoya disease is about 13%, which is significantly lower than that in Asians [71]. It is also unclear whether cerebral microbleeds are predictors of subsequent hemorrhage in European patients. Vessel wall imaging by three-dimensional high-resolution MRI has been developed, and it has been suggested that it is related to the hemorrhagic onset as well as ischemic onset. The enhancement of the intracranial vessel wall is associated with ipsilateral initial bleeding, suggesting that vessel wall enhancement may be a useful marker of hemorrhagic onset [72].

Abovementioned, the presence of intracranial aneurysms is also associated with the risk of bleeding, for example as intracerebral or subarachnoid hemorrhage. Although there are a few reports of aneurysms associated with moyamoya disease, the predominant site of aneurysms in such cases was around the circle of Willis (56%), followed by the basal ganglia, collateral vessels, and other vessels [73]. Follow-up angiography of the aneurysms showed that all aneurysms in the basal ganglia or collateral vessels resolved without surgical intervention [73], which means that surgical intervention for aneurysms may need to be recommended to prevent the rupture of aneurysms in the circle of Willis. Clinical factors associated with hemorrhagic onset moyamoya disease are summarized in a table (Table 4).

## 7. Factors Associated with the Recurrence of Hemorrhagic Events in Hemorrhagic Onset Moyamoya Disease

In cases of hemorrhagic moyamoya disease, rebleeding is also an important factor in worsening prognosis, reducing recovery rates, and increasing the mortality rate [55]. When the course is followed by conservative treatment, the probability of a rebleeding event occurring after 2–20 years is 28–61% [54,55,74,75]. In a randomized controlled trial called the Japan Adult Moyamoya Trial, the rebleeding rate for hemorrhagic moyamoya disease in the non-surgical group was 7.6% per year [76]. As rebleeding may occur after a prolonged interval (10 years or more), long-term preventive measures for rebleeding are needed. As mentioned above, the same type of stroke as that at the time of initial presentation is likely to occur in adult moyamoya patients. In a retrospective study on 241 patients with moyamoya disease, multivariate analysis of risk factors revealed that initial hemorrhagic presentation was a significant risk factor for hemorrhagic events [19]. Regarding the clinical features, rebleeding has been reported to be age-related, and the incidence of rebleeding increases when patients reach the age range of 46–55 years [75]. According to a report analyzing the natural history of rebleeding in hemorrhagic moyamoya disease, including imaging features as well as stroke risk factors and family history, smoking is an independent risk factor for rebleeding; on the other hand, age, type of initial bleeding, family history, and digital subtraction angiography staging were not found to be associated with an increased risk of rebleeding [77]. Regarding the site of bleeding, when the patients were divided into anterior hemorrhagic and posterior hemorrhagic groups, rebleeding rates were found to be significantly higher in the posterior bleeding group than in the anterior hemorrhagic group [78]. As already mentioned, choroidal anastomosis and PCA involvement are characteristic of posterior hemorrhage [3], which means that these anatomical features are associated with rebleeding. In fact, choroidal collaterals have been reported to be predictors of rebleeding in patients with hemorrhagic moyamoya disease [79]. Among the collateral choroidal anastomoses, lateral posterior choroidal anastomosis has been reported as the most significant risk factor for hemorrhagic recurrence [80]. The development of choroidal collaterals and stenosis of the PCA are significantly associated with rebleeding, possibly because the PCA supplies blood flow anteriorly through leptomeningeal anastomosis, and the presence of stenosis of the PCA results in a hemodynamic overload on collateral circulation, such as choroidal anastomosis. Another study has reported that the presence of intraventricular hemorrhage is a risk factor for recurrent hemorrhage [81]. Cortical hemodynamic failure was also reported to be a risk factor for rebleeding, and a decrease in CVR on SPECT is as an independent risk factor for subsequent hemorrhage in adult hemorrhagic moyamoya disease [82]. Factors associated with recurrence of hemorrhagic symptoms in hemorrhagic onset moyamoya disease are summarized in a table (Table 5).

## 8. Asymptomatic Moyamoya Disease

Asymptomatic moyamoya disease is defined as a case of the disease characterized by the absence of previous cerebrovascular events such as TIA, cerebral infarction, or intracranial hemorrhage [83]. With the development of non-invasive diagnostic methods such as MRI/magnetic resonance angiography (MRI/A), the chances of being diagnosed with moyamoya disease before symptom onset are increasing. Epidemiologically, in a 1994 survey, asymptomatic moyamoya disease accounted for approximately 1.5% of the total number of patients [84]; however, this was found to have increased to 17.8% in a 2008 survey [6]. However, the precise statistics regarding the prevalence and incidence of asymptomatic moyamoya disease are unknown. One estimates of the prevalence of moyamoya disease in the Japanese population based on MRI/A was 50.7 per 100,000 people, which is much higher than expected [85]. It is estimated that there are many patients with moyamoya disease who have not yet been diagnosed, and the prevalence is expected to increase further in the future.

Little is known about the natural history of asymptomatic moyamoya disease. In asymptomatic cases, cerebral infarction and disturbed cerebral hemodynamics were detected in approximately 20% and 40% of the involved hemispheres, respectively [86]. The angiographic stage was generally more advanced in more elderly patients [86]. In a study that analyzed 10 patients with asymptomatic moyamoya disease (average follow-up period, 4.1 years), only one patient presented with a stroke [87]. In a 2007 study by Kuroda et al. that analyzed 40 patients with asymptomatic moyamoya disease, of the 34 patients treated non-surgically (average follow-up period, 43.7 months), three patients had TIA, one patient had cerebral infarction, and three patients presented with intracranial bleeding [86]. According to another study that retrospectively analyzed 84 cases of asymptomatic moyamoya disease who were followed from the time of diagnosis (mean observation period, 42.2 months), 7 patients became symptomatic (cerebral infarction, *n* = 2; TIA, *n* = 5; hemorrhage, *n* = 0) [4]. Asymptomatic moyamoya disease not treated surgically has an annual incidence of cerebrovascular events, including TIA, of 2.4–5.7% [4,87,88].

There are a few reports on the risk factors for cerebrovascular events in moyamoya disease. One retrospective analysis of the natural history of 84 cases of asymptomatic moyamoya disease reported that stroke risk factors, such as hypertension and dyslipidemia, can increase the risk of cerebrovascular events by 6.7–8.1 times [4]. Smoking has also been reported to be associated with the development of TIA-related symptoms in asymptomatic patients [89]. As for radiological features, decreased CVRs indicate progression of the disease stage and that symptoms should be carefully monitored on a regular basis [89,90]. The presence of microbleeds on MRI is a significant predictor of hemorrhagic stroke in adult patients with moyamoya disease [91]. The natural history of normal hemispheres and the hemispheres that underwent surgery in patients with unilateral moyamoya disease has also been investigated. According to cohort studies on non-surgical hemispheres, imaging stage progression occurred in approximately 24% of patients during the follow-up period (mean follow-up period, 73.6 months); moreover, about half of the patients with advanced disease developed ischemic or hemorrhagic events [92]. In another study, which was, however, not strictly a study of asymptomatic moyamoya disease, the presence of choroidal collaterals was found to raise the risk of de novo hemorrhage in the unaffected non-hemorrhagic hemisphere [93]. In addition, unilateral moyamoya disease has also been reported to shift bilaterally [17,94,95]. Sixty-five percent of children with moyamoya disease with unilateral ischemic symptoms had TIAs on the asymptomatic side [17]. Children or young adults with unilateral moyamoya disease were found to be likely to develop bilateral disease within 1–5 years [94]. The presence of minor changes in the contralateral intracranial major arteries is a predictor of increased risk of progression [95,96]. In summary, asymptomatic moyamoya disease can cause cerebrovascular events, including ischemia and hemorrhage; thus, follow-up using MRI/A is recommended, as the identification of high-risk groups may improve the future outcomes. The AMORE study (NCT02687828), a multicenter randomized study on asymptomatic moyamoya disease, is currently in progress [83], and the results are awaited. Factors associated with cerebrovascular events in asymptomatic moyamoya disease are summarized in a table (Table 6).

## 9. Summary of the Literature Review

So far, we have discussed the association between the onset pattern and clinical features. The results are classified into background features and radial features and summarized in a table (Table 7).

## 10. Conclusions

We have summarized the findings of the studies conducted to date on the clinical features of the onset pattern and natural history of moyamoya disease. From our review, since each onset pattern has its own imaging characteristics, such as ischemic onset and PCA involvement or blood flow decrease in the blood flow test, hemorrhagic onset and periventricular anastomosis or microbleeds, it is important to carefully assess the images in preoperative examinations and follow-ups and to provide appropriate therapeutic intervention at the appropriate time. Furthermore, since stroke risk factors, such as hypertension, dyslipidemia, and smoking, are an important risk in the symptomization of asymptomatic moyamoya disease, proper medical management in asymptomatic cases can also lead to prevention of future cerebrovascular events. However, since many aspects of the etiology and natural history of this disease remain unclear, it is essential to conduct a larger prospective study in the future.

## Figures and Tables

**Table 1 jcm-10-02815-t001:** Clinical factors associated with ischemic onset moyamoya disease.

Factors	Author, Year	Sample Size (Population)	Details
**Background Features**			
Age < 6 years	Kim et al., 2004 [13]	204 cases (Korea)	At initial presentation, infarctions were significantly more frequent in children <3 years (87%) of age and those 3–6 years of age (58%) than in those 6 years or older (46%).
Female sex (pediatric)	Zhao et al., 2017 [18]	60 pediatric cases (China)	Female gender was an independent predictor of future ischemic stroke (HR, 5.08 (1.40–18.47); *p* = 0.01).
Idiopathic moyamoya (pediatric)	Dlamini et al., 2020 [37]	37 pediatric cases (Canada)	Children with idiopathic moyamoya were at significantly greater risk of ischemic events (HR, 3.71 (1.1–12.8); *p* = 0.037).
Familial moyamoya	Cho et al., 2015 [19]	241 adult cases (Korea)	Familial moyamoya disease was a significant risk factor for an ischemic event (HR, 3.108 (1.260–7.665); *p* = 0.014).
*RNF213* c.14429G>A (p.Arg4810Lys, rs112735431) variant	Miyatake et al., 2012 [22]	204 cases (Japan)	Homozygous *RNF213* c.14429G>A (AA) was associated with • Earlier onset (vs. heterozygote (GA): *p* = 0.002, vs. wild type (GG): *p* = 0.007). • An increased risk of cerebral ischemia at initial presentation (vs. heterozygote (GA): *p* = 0.01, vs. wild type (GG): *p* = 0.01).
	Kim et al., 2016 [23]	165 cases (Korea)	Compared to those with GG, patients with *RNF213* GA or AA were more likely to experience: • Early onset MMD (*p* = 0.001). • Cerebral infarction at presentation (*p* = 0.002).
	Wang et al., 2021 [28]	2798 cases (China, Korea, Japan)	• More patients were aged <15 years in the GA and AA groups (AA vs. GA: *p* = 0.009; AA vs. GG: *p* = 0.003; GA vs. GG: *p* = 0.001). • More patients were aged <4 years in the AA group (AA vs. GA: *p* < 0.00001; AA vs. GG: *p* < 0.00001). • More infarctions in AA group (AA vs. GA: *p* < 0.004; AA vs. GG: *p* < 0.004).
Hyperthyroidism and elevated TPOAb	Ahn et al., 2018 [30]	169 cases (Korea)	Hyperthyroidism had an increased risk of MMD with ischemic stroke with reference value of MMD without stroke (OR, 2.53; *p* = 0.055). Anti-thyroperoxidase antibody (TPOAb) increased the risk of MMD presenting with ischemic stroke significantly (OR, 2.99; *p* = 0.020).
**Radiological Features**			
Advanced Suzuki’s angiographical stage (pediatric)	Zhao et al., 2017 [18]	60 pediatric cases (China)	Suzuki’s angiographical stage greater than 3 was an independent predictor of future ischemic stroke (HR, 4.01 (1.16–13.82); *p* = 0.03).
PCA involvement	Ohkura et al., 2018 [32]	93 cases (Japan)	PCA steno-occlusive lesions were significantly more frequent in the infarcted (77.8%) than in non-infarcted hemispheres.
Advancement of PCA stages (adult)	Zhao et al., 2018 [33]	574 cases (China)	The frequency of ipsilateral cerebral infarction was significantly positively correlated with the advancement of PCA stages in adult patients (*p* < 0.001), but not in pediatric patients (*p* = 0.106).
Steno-occlusive lesions at the proximal ICA	Ohkura et al., 2018 [32]	93 cases (Japan)	Steno-occlusive lesions at the ICA proximal to the PCoA were significantly more frequent in infarcted (27.8%) than in non-infarcted hemispheres.
Contrast enhancement and thickening of the vessel wall	Kathuveetil et al., 2020 [35]	29 cases (India)	The presence of contrast enhancement (*p* = 0.01) and wall thickening (*p* ≤ 0.001) showed a statistically significant association with ischemic events.
Relative value of CBV and prolongation of the MTT	Hirai et al., 2017 [36]	122 cases (Japan)	The relative value of CBV (*p* < 0.01) and prolongation of the MTT in comparison with the cerebellum (*p* < 0.05) was significantly higher in the TIA and ischemic onset group than in the asymptomatic group.
Presence of steal on blood oxygen level-dependent MRI (pediatric)	Dlamini et al., 2020 [37]	37 pediatric cases (Canada)	The presence of steal was independently associated with ischemic events (OR, 19.8 (2.5–160); *p* = 0.005).
CBF ratio of the lenticular nucleus to the MCA territory on SPECT	Arai et al., 2020 [39]	85 cases (Japan)	The ratio of CBF in one lenticular nucleus to that in the peripheral MCA obtained by SPECT at rest was significantly higher in hemispheres with ischemic symptoms than in those without symptoms (*p* < 0.001).
Microembolic signals detected by transcranial Doppler	Chen et al., 2014 [40]	54 cases (China)	The presence of microembolic signals detected by transcranial Doppler ultrasonography was associated with new ischemic stroke (HR, 10.61 (1.66–67.70); *p* = 0.012).
	Jeon et al., 2019 [41]	48 adult cases (Korea)	Presence of MESs was associated with recent ischemic events (*p* = 0.024).

CBF: cerebral blood flow, CBV: cerebral blood volume, HR: hazard ratio, ICA: internal carotid artery, MCA: middle cerebral artery, MES: microembolic signal, MMD: moyamoya disease, MRI: magnetic resonance imaging, MTT: mean transit time, OR: odds ratio, PCA: posterior cerebral artery, PCoA: posterior communicating artery, RNF: RING finger, SPECT: single-photon emission computed tomography, TIA: transient ischemic attack, TPOAb: Anti-thyroperoxidase antibody.

**Table 2 jcm-10-02815-t002:** Factors associated with recurrence of ischemic symptoms in ischemic onset moyamoya disease.

Factors	Author, Year	Sample Size (Population)	Details
**Background features**			
Initial ischemic presentation	Cho et al., 2015 [19]	241 adult cases (Korea)	Initial ischemic presentation was a significant risk factor for an ischemic event (HR, 2.686 (1.150–6.272); *p* = 0.022).
Age < 3 years	Kim et al., 2004 [13]	204 cases (Korea)	Subsequent infarctions occurred significantly more frequently in children <3 years of age (39%) than in those 3–6 years of age (6%) and 6 years or older (0.8%).
DM (adult)	Noh et al., 2015 [42]	104 adult cases (Korea)	DM was an independent predictor of recurrence of ischemic stroke (HR, 35.16 (2.61–474.16); *p* = 0.007).
**Radiological features**			
PCA involvement (adult)	Noh et al., 2015 [42]	104 adult cases (Korea)	Presence of steno-occlusive lesions in the PCA was an independent predictor of recurrence of ischemic stroke (HR, 17.53, (2.02–152.43); *p* = 0.009).
Decreased CVR (adult)	Noh et al., 2015 [42]	104 adult cases (Korea)	Decreased CVR was an independent predictor of recurrence of ischemic stroke (HR, 13.62 (1.55–119.84); *p* = 0.019).
Ivy sign (adult)	Nam et al., 2019 [44]	84 adult cases (Korea)	An ipsilateral ivy sign was an independent predictor of 3-year ischemic recurrence (adjusted HR, 10.15 (2.10–49.14); *p* = 0.004).

CVR: cerebrovascular reactivity, DM: diabetes mellitus, HR: hazard ratio, PCA: posterior cerebral artery.

**Table 3 jcm-10-02815-t003:** Factors associated with hemorrhagic transformation in ischemic onset moyamoya disease.

Factors	Author, Year	Sample Size (Population)	Details
**Background features**			
Age > 20 years, more than 10 years after bypass surgery	Funaki et al., 2014 [46]	56 pediatric-onset cases (Japan)	Three of the 56 patients developed late hemorrhage, all in the mid to late 20s, more than 10 years after the previous bypass surgery.
**Radiological features**			
Normal cerebral perfusion	Lu et al., 2020 [45]	683 cases (China)	Normal cerebral perfusion according to CT was associated with hemorrhagic transformation (OR, 13.46 (3.53–51.36); *p* < 0.001).

CT: computed tomography, MMD: moyamoya disease, OR: odds ratio.

**Table 4 jcm-10-02815-t004:** Clinical factors associated with hemorrhagic onset moyamoya disease.

Factors	Author, Year	Sample Size (Population)	Details
**Background features**			
Onset age	Yamamoto et al., 2020 [57]	69 adult cases (Japan)	Childhood onset adult moyamoya disease was significantly associated with the occurrence of hemorrhagic stroke compared to adulthood onset adult moyamoya disease (OR = 4.31 (1.21–15.4); *p* = 0.025).
	Zhang et al., 2020 [59]	335 cases (China)	Age at onset was negatively associated with a significantly increased risk of posterior hemorrhage (OR = 0.98 (0.96–1.00); *p* = 0.048).
HT	Zhang et al., 2020 [59]	335 cases (China)	HT was a risk factor for anterior hemorrhage in patients without dilation of choroidal anastomosis (OR 0.37; (0.14–0.97); *p* = 0.043).
Familial moyamoya	Cho et al., 2015 [19]	241 adult cases (Korea)	Familial moyamoya disease was a significant risk factor for a hemorrhagic event (HR, 3.094 (1.238–7.730); *p* = 0.016).
**Radiological features**			
Advanced Suzuki’s angiographical stage	Fujimura et al., 2019 [60]	155 cases (Japan)	Suzuki’s angiographical stage was significantly higher in the hemorrhagic onset group than in the ischemic onset group (*p* = 0.038).
	Jang et al., 2014 [61]	175 cases (Korea)	Suzuki’s angiographical stage had a strong tendency to be more advanced in hemorrhagic than in ischemic patients (*p* = 0.061).
	Yamamoto et al., 2019 [62]	110 cases (Japan)	Suzuki’s angiographical stage was more advanced in the hemorrhagic group than in the asymptomatic group (*p* = 0.0033).
	Gross et al., 2013 [63]	42 cases (United States)	The mean Suzuki’s angiographical stage was higher in patients presenting with hemorrhage (3.7 compared to 2.9, *p* = 0.03)
Occlusions of the ACA	Jang et al., 2014 [61]	175 cases (Korea)	Occlusions of the ACA were more frequently observed in the hemorrhagic than the ischemic (*p* = 0.001) or control hemispheres (*p* = 0.011).
Fetal-type PCA	Jang et al., 2014 [61]	175 cases (Korea)	Development of fetal-type PCA was more frequently observed in the hemorrhagic than the ischemic (*p* = 0.01) or control hemispheres (*p* = 0.013).
Intracranial aneurysms	Jang et al., 2014 [61]	175 cases (Korea)	Intracranial aneurysms were more frequently found in the hemorrhagic than the ischemic or control hemispheres (*p* = 0.002).
Dilatation and branch extension of AChA, PChA, PCoA, and medullary arteries on angiography or MRA (especially adult)	Irikura et al., 1996 [64]	19 adult cases (Japan)	Marked enlargement of the choroidal arteries and the medullary arteries was seen more frequently in the hemorrhagic group.
	Morioka et al., 2003 [65]	107 cases (Japan)	In adult patients, the proportion of dilation and abnormal branching of the anterior choroidal artery and posterior communicating artery was significantly higher in hemorrhagic hemispheres than in the ischemic and asymptomatic hemispheres (*p* < 0.01).
	Liu et al., 2011 [66]	132 adult cases (China)	Extension with abnormal branches and excessive dilation of the AChA-PCoA accounted for 28 of the hemorrhagic lesions (43.8%), especially intraventricular hemorrhage (57.1%; *p* < 0.001).
	Yamamoto et al., 2018 [67]	41 cases (Japan)	The PCoA, AChA, and PChA more distinctly developed in hemispheres with intracerebral or intraventricular hemorrhage than in hemispheres with ischemic stroke or a transient ischemic attack (*p* < 0.001, *p* = 0.03, and *p* = 0.03, respectively).
	Hirano et al., 2020 [4]	178 cases (Japan)	The proportion of patients who developed AChA was significantly higher in the hemorrhagic group than in the asymptomatic group (*p* = 0.0042) or the ischemic group (*p* < 0.0001).
Periventricular anastomosis detected with MRA	Funaki et al., 2016 [68]	122 cases (Japan)	Periventricular anastomosis score was a factor tentatively associated with hemorrhagic presentation (*p* < 0.01).
Thalamic anastomosis	Fujimura et al., 2019 [60]	155 cases (Japan)	Hemorrhagic-onset patients showed a significantly higher proportion of thalamic anastomosis (*p* = 0.043) compared with ischemic-onset patients.
	Yamamoto et al., 2019 [62]	110 cases (Japan)	The development of thalamic anastomosis was more pronounced in the hemorrhagic group than that in the asymptomatic group (*p* = 0.011).
Choroidal anastomosis	Funaki et al., 2018 [3]	75 cases (Japan)	Choroidal anastomosis was a factor associated with posterior hemorrhage (OR, 2.66 (1.00–7.07); *p* = 0.049).
	Fujimura et al., 2019 [60]	155 cases (Japan)	Hemorrhagic-onset patients showed a significantly higher proportion of choroidal anastomosis (*p* < 0.001) compared with ischemic-onset patients.
	Yamamoto et al., 2020 [57]	69 adult cases (Japan)	Development of choroidal channels were significantly associated with the occurrence of hemorrhagic stroke (OR, 6.78 (1.78–25.8); *p* = 0.005).
	Zhang et al., 2020 [59]	335 cases (China)	Choroidal anastomosis was associated with a significantly increased risk of posterior hemorrhage in children and young adults (OR, 2.62 (1.02–6.72); *p* = 0.045).
PCA involvement	Funaki et al. 2018 [3]	75 cases (Japan)	PCA involvement was a factor associated with posterior hemorrhage (OR, 2.92 (1.01–8.46); *p* = 0.049).
	Zhang et al., 2020 [59]	335 cases (China)	PCA involvement was associated with a significantly increased risk of posterior hemorrhage in children and young adults (OR, 3.39 (1.20–9.63); *p* = 0.022).
	Yamamoto et al., 2019 [62]	110 cases (Japan)	The prevalence of PCA involvement was significantly higher in the hemorrhagic group than in the asymptomatic group (*p* = 0.016).
Microbleeds	Kikuta et al., 2008 [69]	50 cases (Japan)	The presence of multiple microbleeds might be a predictor of subsequent hemorrhage (HR, 2.89 (1.001–13.24); *p* = 0.0497).
	Sun et al., 2013 [70]	85 cases (China)	Microbleeds in the deep and periventricular white matter were independent predictors of subsequent intraventricular hemorrhage (HR, 5.53 (1.20–25.41); *p* = 0.028).
	Hirano et al., 2020 [4]	178 cases (Japan)	The proportion of patients who developed microbleeds was significantly higher in the hemorrhagic group than in the asymptomatic group (*p* = 0.0008) or the ischemic group (*p* = 0.0002).
Wall enhancement of intracranial vessels	Lu et al., 2021 [72]	507 cases (China)	Vessel wall enhancements were independently associated with ipsilateral initial hemorrhage (OR, 1.99 (1.20–3.28); *p* = 0.007).

ACA: anterior cerebral artery, AChA: anterior choroidal artery, HR: hazard ratio, HT: hypertension, MRA: magnetic resonance angiography, OR: odds ratio, PCA: posterior cerebral artery, PChA: posterior choroidal artery, PCoA: posterior communicating artery.

**Table 5 jcm-10-02815-t005:** Factors associated with recurrence of hemorrhagic symptoms in hemorrhagic onset moyamoya disease.

Factors	Author, Year	Sample Size (Population)	Details
**Background features**			
Initial hemorrhagic presentation	Cho et al., 2015 [19]	241 adult cases (Korea)	Initial hemorrhagic presentation was a significant risk factor for a hemorrhagic event (HR, 2.527 (1.236–5.166), *p* = 0.011).
Age 46–55 years	Morioka et al., 2003 [75]	36 cases (Japan)	Rebleeding occurs at an increased rate when patients reach the age range of 46 to 55 years.
Smoking	Kang et al., 2019 [77]	128 cases (China)	Smoking was an independent risk factor for rebleeding (OR, 4.85; *p* = 0.04).
**Radiological features**			
Posterior hemorrhage	Takahashi JC et al., 2016 [78]	80 cases (Japan)	Analysis within the nonsurgical group revealed that the incidence of rebleeding was significantly higher in the posterior group than in the anterior group.
Choroidal collaterals (adult)	Funaki et al., 2019 [79]	37 adult cases (Japan)	The presence of choroidal collaterals was a significant predictor of rebleeding in hemorrhagic moyamoya disease (adjusted HR, 11.10 (1.37–89.91)).
Lateral posterior choroidal anastomosis (adult)	Wang et al., 2019 [80]	37 adult cases (China)	Lateral posterior choroidal artery anastomosis was associated with recurrent hemorrhage (HR, 5.78 (1.58–21.13); *p* < 0.01).
Intraventricular hemorrhage (adult)	Kim et al., 2017 [81]	176 cases (Korea)	The presence of intraventricular hemorrhage had a marginal significance for recurrent hemorrhage (HR, 3.32, *p* = 0.05).
Decreased CVR (adult)	Takahashi JC et al., 2020 [82]	79 adult cases (Japan)	Hemodynamic failure with decreased CVR was an independent risk factor for subsequent hemorrhage in hemorrhagic moyamoya disease (HR, 5.37 (1.07–27.02); *p* < 0.05).

CVR: cerebrovascular reserve, HR: hazard ratio, OR: odds ratio.

**Table 6 jcm-10-02815-t006:** Factors associated with cerebrovascular events in asymptomatic moyamoya disease.

Factors	Author, Year	Sample Size (Population)	Details
**Background features**			
HT	Hirano et al., 2020 [4]	84 cases (Japan)	HT may increase the risk of cerebrovascular events in asymptomatic patients (HR 6.69 (1.23–36.4); *p* = 0.028).
DL	Hirano et al., 2020 [4]	84 cases (Japan)	DL may increase the risk of cerebrovascular events in asymptomatic patients (HR 8.14 (1.46–45.2); *p* = 0.017).
Smoking (adult)	Jo et al., 2014 [89]	40 adult cases (Korea)	Transient ischemic attack was associated with smoking in the non-surgical group (*p* = 0.017).
**Radiological features**			
Decreased CVR (adult)	Jo et al., 2014 [89]	40 adult cases (Korea)	Transient ischemic attack was associated with decreased vascular reserve (*p* < 0.001) in the non-surgical group in asymptomatic moyamoya disease.
	Yang et al., 2014 [90]	42 adult cases (Korea)	Disease progression in asymptomatic moyamoya disease was associated with initial cerebrovascular reserve capacity (*p* = 0.05).
Microbleeds (adult)	Kuroda et al., 2013 [91]	78 adult cases (Japan)	The presence of silent microbleeds was a significant predictor for subsequent hemorrhagic stroke in adult moyamoya disease (*p* < 0.001).
Choroidal anastomosis (adult)	Funaki et al., 2019 [93]	36 cases (Japan)	The annual risk of de novo hemorrhage in non-hemorrhagic hemispheres was significantly higher in the collateral-positive group than in the collateral-negative group (5.8% per year vs. 0% per year; *p* = 0.017).

CVR: cerebrovascular reactivity, DL: dyslipidemia, HR: hazard ratio, HT: hypertension, MMD: moyamoya disease.

**Table 7 jcm-10-02815-t007:** Summary of clinical factors of clinical features associated with each onset pattern in moyamoya disease.

	Background Features	Radiological Features
Clinical factors associated with ischemic onset	Age < 6 years Female sex (pediatric) Idiopathic moyamoya (pediatric) Familial moyamoya *RNF213* c.14429G>A (p.Arg4810Lys, rs112735431) variant Hyperthyroidism and elevated TPOAb	Advanced Suzuki’s angiographical stage (pediatric) PCA involvement Advancement of PCA stages (adult) Steno-occlusive lesions at the proximal ICA Contrast enhancement and thickening of the vessel wall Relative value of CBV and prolongation of the MTT Presence of steal on blood oxygen level-dependent MRI (pediatric) CBF ratio of the lenticular nucleus to the MCA territory on SPECT Microembolic signals detected by transcranial Doppler
Clinical factors associated with ischemic recurrence	Initial ischemic presentation Age < 3 years DM (adult)	PCA involvement (adult) Decreased CVR (adult) Ivy sign (adult)
Clinical factors associated with hemorrhagic transformation	Age > 20 years, more than 10 years after bypass surgery	Normal cerebral perfusion
Clinical factors associated with hemorrhagic onset	Onset age HT Familial moyamoya	Advanced Suzuki’s angiographical stage Occlusions of the ACA Fetal-type PCA Intracranial aneurysms Dilatation and branch extension of AChA, PChA, PCoA, and medullary arteries on angiography or MRA (especially adult) Thalamic anastomosis Choroidal anastomosis PCA involvement Microbleeds Wall enhancement of intracranial vessels
Clinical factors associated with hemorrhagic recurrence	Initial hemorrhagic presentation Age 46–55 years Smoking	Posterior hemorrhage Choroidal collaterals (adult) Lateral posterior choroidal anastomosis (adult) Intraventricular hemorrhage (adult) Decreased CVR (adult)
Clinical factors associated with cerebrovascular events in asymptomatic moyamoya disease	HT DL Smoking (adult)	Decreased CVR (adult) Microbleeds (adult) Choroidal anastomosis (adult)

ACA: anterior cerebral artery, AChA: anterior choroidal artery, CBF: cerebral blood flow, CBV: cerebral blood volume, CVR: cerebrovascular reactivity, DL: dyslipidemia, DM: diabetes mellitus, HR: hazard ratio, HT: hypertension, ICA: internal carotid artery, MCA: middle cerebral artery, MMD: moyamoya disease, MRA: magnetic resonance angiography, MRI: magnetic resonance imaging, MTT: mean transit time, PCA: posterior cerebral artery, PChA: posterior choroidal artery, PCoA; posterior communicating artery, SPECT: single-photon emission computed tomography, TPOAb: Anti-thyroperoxidase antibody.

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
