# Peer review of "Differences in Clinical Features among Different Onset Patterns in Moyamoya Disease"

_jcm, 2021, doi:10.3390/jcm10132815_

Round 1
Reviewer 1 Report
Very useful overview on the risk factors for ischemic and hemorrhagic manifestations. Unfortunately, no studies from the USA or Europe have been evaluated. What are the differences between Asian and Caucasian patients?
A risk score for (recurring) ischemia or intracranial bleeding would be helpful.
Reviewer 2 Report
Great effort.Many data in an extremely complicated way.You are not mentioned the method of selection of these studies and your purpose of this review is unclear.It would be better to state your conclusion regarding these retrospective studies.
In Table 2 i believe you mean ischemic event and not hemorrhagic by mistake(1st line).
Reviewer 3 Report
The authors of the manuscript “Differences in clinical features among different onset patterns in moyamoya disease” present a summary of the literature on the body of evidence available for the clinical features associated with different onset patterns of moyamoya disease. A narrative review was written.
The manuscript is well structured and written in good English. The different onset patterns are adequately discussed and the manuscript seems to summarize the current state of knowledge. A more elaborate presentation of the large clinical studies, with their respective results, would make the manuscript even more interesting and strengthen the results.
Major comments:
- It is not explained how the articles for this review were selected. As it is a review of the literature, the authors need to explain the selection procedure.
- The authors should use the PRISMA checklist to organize the points mentioned above.
- I advise to additionally create a bullet-point overview of the main features per onset pattern to make it more easy to remember.
Below are some other minor comments:
- The mentioning of the genetic component in the introduction suggests that he review also includes this topic. However, no further action is taken to explain the importance of ring finger protein 213. I would advise to remove this part.
- Please adjust the typo in line 83-84: …the age of 10 years of age…
- Several times, the authors mention ‘a difference’ between groups (e.g. line 141). I would recommend to report the estimate points of the differences.
- Also, in line 147-148, it is not clear which ratio is relevant. At which ratio is there an increased risk?
- The line 197 is somewhat awkward, because this review tries to give an overview of literature, but also mentions that current studies are being done. Why are those studies not including in this manuscript?
- The final sentence does not reflect the results of this review, especially regarding the genetic factors. Does more knowledge about genetic factors improve the understanding?
Round 2
Reviewer 1 Report
Improved manuscript
Reviewer 3 Report
Thank you for improving your manuscript and addressing all of my comments.